# Numerical Analysis of Aeroacoustic Phenomena Generated by Truck Platoons

**Władysław Marek Hamiga \* and Wojciech Bronisław Ciesielka \***

Department of Power Systems and Environmental Protection Facilities, AGH University of Science and Technology, 30-059 Kraków, Poland

\* Correspondence: hamiga@agh.edu.pl (W.M.H.); ghciesie@cyf-kr.edu.pl (W.B.C.)

**Abstract:** In recent years there has been dynamic progress in the development of fully autonomous trucks and their combination and coordination into sets of vehicles moving behind each other within short distances, i.e., platooning. Numerous reports from around the world present significant benefits of platooning for the environment due to reduced emissions, reduced fuel costs, and improved logistics in the transport industry. This paper presents original aerodynamic and aeroacoustic studies of identical truck column models. They are divided into four main stages. In the first, a truck model and three columns of identical trucks with different distances between the vehicles was made and tested using computational fluid dynamics (CFD). Two turbulence models were used in the study: $k - \omega$ shear stress transport (SST) and large eddy simulation (LES). The aim of the work was to determine the drag coefficients for each set of vehicles. The second stage of work included determination of sound field distributions generated by moving vehicles. Using the Ffowcs Williams–Hawkings (FW-H) analogy, the sound pressure levels were determined, followed by the sound pressure levels A. In order to verify the correctness of the work carried out, field tests were also performed and additional acoustic calculations were carried out using the NMPB-Routes-2008 and ISO 9613-2 models. Calculations were performed using SoundPlan software. The performed tests showed good quality of the built aerodynamic and aeroacoustic models. The results presented in this paper have a universal character and can be used to build intelligent transport systems (ITSs) and intelligent environmental management systems (IEMSs) for municipalities, counties, cities, and urban agglomerations by taking into account the platooning process.

**Keywords:** aerodynamics; aeroacoustics; truck platoon; sustainability; LES; CFD; CAA; FW-H; ITS; IEMS

## 1. Introduction

Road freight transport is one of the most important transport branches in the world. The development of road transport is closely related to the existence of the necessary road infrastructure and the systematic development of the road fleet. The basic task of road freight transport is to carry goods from the place of shipment to the destination using specific means of transport. An indispensable condition for road transport to function is the existence of transport infrastructure, both point infrastructure, i.e., facilities for stationary handling of means of transport and cargo, such as reloading yards and points, public unloading facilities, or logistic centers, and line infrastructure, i.e., various types of roads and highways. The implementation of modern solutions in trucking, such as truck platooning, robotization of processes, and the use of alternative drives, has been driving the transportation industry very dynamically over the past two decades. Their implementation is supposed to improve transport efficiency and safety and minimize harmful influence on the environment. The authors of the report [1], prepared by the Polish Economic Institute together with the Ministry of Infrastructure, indicate that by 2040 automation will make it possible to significantly increase the comfort of drivers' work while reducing labor costs by 10 percent. The report also indicates that, thanks to collating convoys from trucks, the

environment will also benefit by reducing fuel and material consumption in transport by 5 percent and $CO_2$ emissions by 10 percent.

Platooning is a term related to the way of moving two or more vehicles in convoy, on the principle of trains. A group of vehicles that use the mutual communication system move in specific columns. The communication system, e.g., vehicle to vehicle system (V2V), allows the vehicles in the convoy to maintain a strictly defined spacing between the cars. This task is accomplished mainly by simultaneous acceleration and braking of all vehicles.

The main advantages of platooning include reducing the amount of fuel consumed, reducing carbon dioxide emissions, improving road safety, and improving the comfort and smoothness of traffic. Due to the presented advantages, many international projects using the idea of platooning have been created. The programs are implemented with the support of global transport concerns, such as: Scania, Iveco, MAN, Volvo, DAF, or Mercedes. Examples include projects such as: Safe Road Trains for the Environment (SARTRE) [2,3], PATH [3,4], or Grand Cooperative Driving Challenge (GCDC) [3] for research on heterogeneous columns, or KONVOI [5], Energy ITS [3], SCANIA [3], COMPANION [6], and European Truck Platoon Challenge (ETPC), related to homogeneous columns, which only include trucks.

These studies, which concern the analysis of the air drag forces acting on the moving vehicles in the column, have been carried out for years. In 1994, Zabat et al. [7] presented studies on the use of aerodynamic wake in columns of two, three, and four vehicles.

Lammert et al. [8], in 2014, conducted tests under normal operating conditions for two trucks. The investigation included different spacing from 6 to 22.5 m and influence of the speed of the moving column on the reduction of fuel consumption.

In 2016, Humphreys et al. [9] also tested a homogeneous column consisting of two trucks. However, this time, the influence on the generated air drag force of a 2 ft lateral offset of one of the cars was checked.

Studies on columns of vehicles were carried out by Siemon [10] in 2018 on a homogeneous column consisting of four trucks and a heterogeneous column consisting of four trucks. The difference between the vehicles resulted from the type of load on the trailer. Vohra [11] studied homogeneous columns consisting of two and three trucks.

Jaffar [12] used machine learning to determine the drag force coefficient of a vehicle column. The objects of the research were homogeneous columns consisting of two, three, and four vehicles. The study was carried out with five training algorithms: linear regression, polynominal regression, support vector regression, and two models of neural networks. The experimental research conducted in [13] was used as a set of learning data. Additionally, the authors performed CFD simulations for analogous homogeneous columns. The results prove the usefulness of supervised learning methods as an additional tool supporting the analysis of aerodynamic parameters of vehicles.

In the work of Kaluva [14], the air drag coefficient was analyzed for two types of homogeneous columns. The first column consists of the dynamic autonomous road transit (DART), while the second one consists of passenger notchback vehicles. The aerodynamic parameters were checked for each vehicle separately as well as for columns built from two to seven vehicles. The research used ANSYS Fluent, software designed to perform CFD simulations. The discretization of the model was performed using a non-structural mesh. The averaged air drag coefficient was reduced to 23% for the DART vehicle column and 24% for the passenger vehicle column.

Robertson et al. [15] described experimental studies carried out for homogeneous columns. The trucks, made on a scale of 1/20, constituted a convoy of eight vehicles. The model was characterized by the possibility of imitating the movement of vehicles relative to the ground. Three distances between trucks were tested: 0.5, 1, and 1.5 vehicle lengths at a speed of 25 m per second. For the smallest distance, the reduction in the averaged drag coefficient for each vehicle was 48%.

He et al. [16] conducted simulation analysis for an analogous homogeneous column. The detached eddy simulation (DES) turbulence model was used to analyze the flow and

aerodynamic coefficients of the vehicles. The coherence of CFD tests with the experiment was demonstrated.

The analysis of the implementation of platooning in the near future and the directions of further studies were described by Sivanandham et al. [17]. In addition to presenting the far-reaching benefits of organized road transport, the authors indicate the currently existing limitations that hinder the introduction of new technologies.

Apart from such important elements related to reductions in fuel consumption [18,19] and emissions of harmful substances into the environment [20,21], trucks driving in a column considerably reduce the occupancy of roads [22–24] increasing the throughput [25] and thus increasing traffic safety [26–28]. It should be remembered, however, that excessively long platoons of trucks may devastate the infrastructure, especially bridges and viaducts, and contribute to disruption of traffic flow for other vehicles.

Another direction intensively explored by researchers is related to the use of modern control systems and communication between truck platoon vehicles. Currently, the most widely used is a controller called cooperative adaptive cruise control (CACC). It is an extended version of the adaptive cruise control (ACC). Existing CACC controllers communicate using short range wireless technology [29], while work is underway to use other communication protocols including LTE and 5G [30,31]. The work [32] characterized selected CACC controllers in detail, therefore they will not be discussed extensively in this paper. However, the authors note that numerous new approaches to platoon controller design have been proposed over the past few years, including adaptive [33,34], linear [35], model predictive control [36,37], and sliding mode control [38–42] or consensus-based controller [43,44].

Truck platoons will be important components of intelligent transportation systems (ITSs) [45] and intelligent environmental management systems (IEMSs) [46–50] in the future. Therefore, in this paper, we focus attention on developing and verifying original aerodynamic and aeroacoustic models for creating information layers in ITSs and IEMSs.

The purpose of this work is to create an original, verified, numerical model that allows for determining the acoustic field around a homogeneous truck platoon. Simulations are based on using the large eddy simulation (LES) turbulence model and the Ffwocs Williams–Hawkings analogy, which are implemented in ANSYS Fluent software. Due to the low Mach number, which was Ma = 0.075, the calculations were performed using the Farassant method and presented boundary conditions. The frequency range in this work is 16 kHz. The CFD calculation was performed in the commercial software ANSYS Fluent. The discretization process was realized in the ANSYS ICEM tool, using structural mesh. The developed model will be used for further research on acoustic and aerodynamic phenomena associated with moving road vehicles and will constitute the so-called "active layer" of the Integrated Management System for Acoustic Environment being developed for the capital and royal city of Krakow by the authors of this work [46–50] and intelligent transport systems (ITSs).

## 2. Materials and Methods

### 2.1. Mathematical Model

In this work, the finite volume method is used. The method is implemented in the ANSYS Fluent software. The application allows for solving partial differential equations describing the movement of fluid particles. The basic equations of fluid mechanics are the continuity (1) and momentum (2) equations. To solve these equations, it is necessary to adopt an additional hypothesis related to the viscosity of the fluid. In accordance with previous studies [51], it was decided to use the $k - \omega$ SST turbulence model for the first stage of the research. This model belongs to the Reynolds averaged Navier–Stokes (RANS) family and is used for steady-state calculations. The results are used as an initial condition for transient calculations with the large eddy simulation (LES) turbulence model. Using the Ffowcs Williams–Hawkings Equation (3), the acoustic fields around the trucks

are calculated. For the performed simulations, it is assumed that the fluid is viscous, Newtonian, and incompressible. The effect of gravity is ignored.

Continuity equation:

$$\nabla \cdot u = 0 \tag{1}$$

Momentum equation:

$$\rho_p \frac{du}{dt} = -\nabla p + \nabla \cdot \tau_{ij} \tag{2}$$

where:
$u$—air velocity vector
$p$—pressure
$\rho_p$—air density
$\tau_{ij}$—stress tensor

Ffowcs Williams–Hawkings equation:

$$\frac{1}{c_0^2} \frac{\partial^2 p'}{\partial t^2} - \nabla^2 (\rho') = \frac{\partial^2}{\partial x_i \partial x_j} [T_{ij} H(f)] + \frac{\partial}{\partial t} [Q_n \delta(f)] + \frac{\partial}{\partial x_i} [L_i \delta(f)] \tag{3}$$

where:

$$T_{ij} = \rho u_i u_j + P_{ij} - c_0^2 (\rho - \rho_0) \delta_{ij}$$
$$Q_n = \rho_0 v_n + \rho(u_n - v_n)$$
$$L_i = P_{ij} n_j + \rho u_i (u_n - v_n)$$
$$P_{ij} = p \delta_{ij} - \mu_p \left[ \frac{\partial u_i}{\partial x_j} + \frac{\partial u_j}{\partial x_i} - \frac{2}{3} \frac{\partial u_k}{\partial x_k} \delta_{ij} \right]$$

$u_i$—air velocity component in the $x_i$ direction
$u_n$—air velocity component normal to the surface $f$
$v_i$—surface velocity component in the $x_i$ direction
$v_n$—surface velocity component normal to the surface $f$
$T_{ij}$—Lighthill stress tensor
$P_{ij}$—compressive stress tensor
$H(f)$—Heaviside function
$\delta(f)$—Dirac delta function
$\delta_{ij}$—Kronecker delta
$c_0$—speed of the sound
$\rho'$—density fluctuation.

The presented model is an example of low Mach number flow, where Ma = 0.075. At this Mach number, an incompressible flow is assumed, and therefore a constant speed of sound. Taking this into consideration, it is accepted that $p' = c_0^2 (\rho - \rho_0) = c_0^2 \rho'$. The pressure fluctuation $p'$ is used to calculate the sound pressure level (SPL) $L_{sp}$ according to Equation (4).

$$L_{sp} = 20 log_{10} \frac{p'}{p_{ref}} [dB] \tag{4}$$

where:
$p_{ref} = 2 \times 10^{-5} [Pa]$—reference acoustic pressure
$p'$—pressure fluctuation.

Besides calculations of the acoustic field, the main tested parameter is the drag coefficient $C_d$. The drag coefficient is calculated according to Equation (5) on the basis of the

drag force acting on the vehicles, frontal area *A*, and vehicle speed $u_{veh}$. In this numerical model, vehicle speed is equal to the absolute value of the set airflow velocity *u*.

$$C_d = \frac{2F_d}{\rho_p A |u_{veh}|^2} \tag{5}$$

where:
$F_d$—acting drag force
*A*—frontal area of the vehicle
$u_{veh}$—vehicle speed.

### 2.2. Geometry Model

In the presented work, the aerodynamic and aeroacoustic parameters related to the movement of the fleet of heavy duty trucks are investigated. The indicated quantities are calculated with the use of the finite volume method and the Ffowcs Williams–Hawkings equations implemented in the ANSYS Fluent software. The object of the research is a homogeneous column consisting of three identical trucks. In order to limit the number of elements and avoid low-quality cells, only the main features of the truck body are modeled. The gap between the tractor and the semi-trailer and the fairing above the driver's cabin are taken into account (Figure 1).

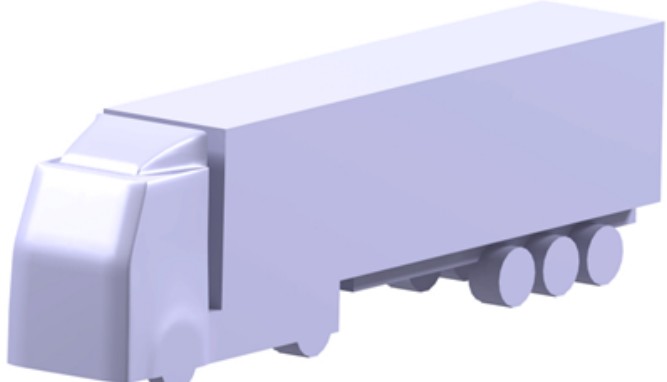

**Figure 1.** Isometric view of truck. CAD model.

The modeling of the chassis details, mirrors, wheel arches, and the exhaust of the vehicles is neglected. The nominal dimensions of the trucks are presented in Figure 2. The length of the trucks is 14 m, the height 4 m, and the width 2.48 m.

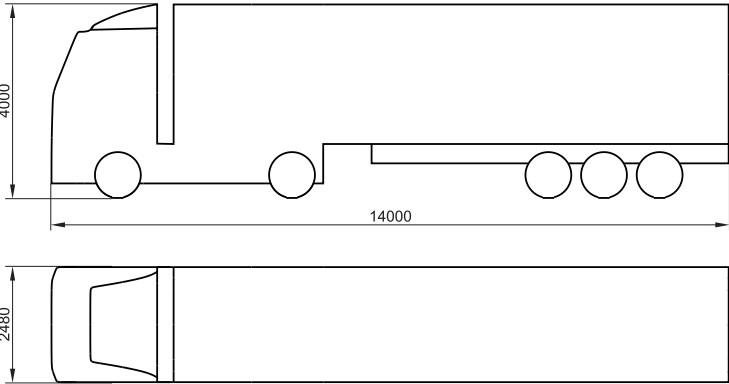

**Figure 2.** Nominal dimensions of the truck. View from side and top.

### 2.3. Discretization

The discretization process is performed in two stages. The first stage is connected with the work publicized by authors in "Aeroacoustic Numerical Analysis of the Vehicle Model" [51]. The article focuses on determining the key parameters of the numerical grid

with respect to comparing the compliance of three criteria: drag coefficient as a function of Reynolds number, streamwise velocity profiles at the symmetry plane, and characteristic structures in the wake of the vehicle model. A validation process is conducted in relation to many research works, experimental as well as numerical. The chosen crucial parameters are connected with modeling of the boundary layer area and quality and type of elements. The second stage is presented in this work. The parameters from the verification process are used for discretization of the truck model domain. In this case, the geometry of the study vehicle is much more complicated and therefore challenging. To achieve the necessary criteria of grid quality, professional software was used with structural mesh. The discretization is performed using the ANSYS ICEM tool, with which the structure of the grid is achieved manually. The boundary layer around the vehicle consists of 15 elements in height. The height of the first element, expressed by the dimensionless y+ factor, does not exceed 1. The growth rate is set to 1.2. A transition layer is used between the boundary layer and the rest of the domain. The transition layer is built analogically to the boundary layer, but it is not in direct contact with the vehicle model. The number of layers in this area is 22 with a growth rate of 1.2.

A schematic of the mesh around the truck model is presented in Figure 3d. The surface mesh on the vehicle is in teal. The first layer is the boundary layer marked in white. The light gray is used to show the transition layer. The dark gray is connected with the discretization of the rest of the domain.

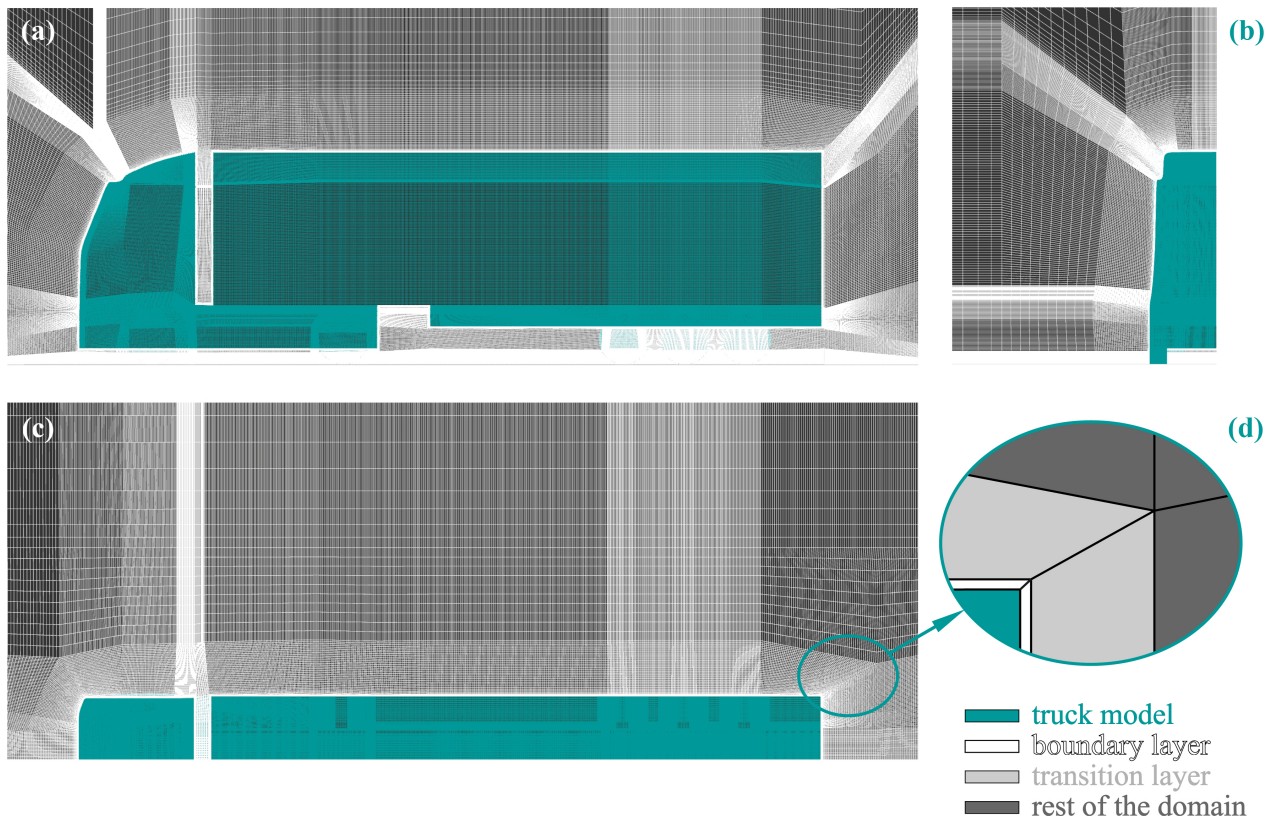

**Figure 3.** Discretization of single truck block. View from: (**a**) side, (**b**) front, and (**c**) top of the tunnel. (**d**) Schematic of grid layers near the boundary.

The developed model is divided into independent blocks. The first block (enter tunnel) is a 25 m lead-in tunnel. This block contains a boundary condition related to the task of air velocity. The next block is a block with the geometry of the tested truck (truck tunnel). Figure 3 presents the discretization of this part of the model. The number of elements of this block is almost 54 million, and the minimum value of the orthogonal quality does not

fall below the value of 0.1. A distance of 2 m was assumed both in front of and behind the vehicle. In order to assemble a model of a column of three vehicles, the block with the vehicle geometry is duplicated twice. For columns where the distance between the vehicles is 8 and 12 m, middle blocks (middle tunnel) with a length of 4 and 8 m, respectively, are added. The model ends with an end block (end tunnel) that is 140 m long. Interfaces are used between the blocks. The grid smoothly transitions from one block to another (conformal interfaces are used). The exact dimensions, number of elements, and the mesh quality of the individual blocks are presented in Table 1.

**Table 1.** Description of mesh blocks.

| Block Name | Block Size [m] | Number of Elements [-] $\cdot 10^6$ | Minimum Orthogonal Quality [-] |
|---|---|---|---|
| Truck tunnel | $18 \times 8 \times 12$ | 53.95 | 0.1 |
| Enter tunnel | $25 \times 8 \times 12$ | 4.09 | 1 |
| Middle tunnel 4 m | $4 \times 8 \times 12$ | 3.29 | 1 |
| Middle tunnel 8 m | $8 \times 8 \times 12$ | 6.50 | 1 |
| End tunnel | $140 \times 8 \times 12$ | 10.51 | 1 |

Mesh Independence Study

In order to check the independence of the obtained results from the adopted mesh density, it was decided to consider additional models. The new models are associated with local refinement and coarsening of the area around the vehicle. The calculations are performed only in the steady state, and the selected parameters and calculation models are applied analogously to those described in this paper. The results of the calculations and the number of model elements are summarized in Table 2.

**Table 2.** Influence of mesh density on aerodynamic parameters.

| Model Name | Total Number of Elements [m] | $C_d$ [-] $\cdot 10^6$ | $C_l$ [-] |
|---|---|---|---|
| Truck_18 | 17.97 | divergence | divergence |
| Truck_63 | 63.33 | 0.572 | −0.127 |
| Truck_345 | 345.47 | 0.574 | −0.132 |

For a model with a coarser mesh within the vehicle model, the calculations are divergent. In the case of a 5 times increase in the number of elements, the differences between the obtained coefficients of the drag force and the coefficients of the lifting force are approx. 0.35% and 3.79%, respectively. Due to the negligible differences in aerodynamic coefficients, it was decided to use the "Truck_63" model. The choice of a model with a much smaller number of elements allows for a more efficient management of computing resources.

*2.4. Boundary Conditions*

In the presented study, three models representing vehicle movement in homogeneous columns are developed. The columns consist of three identical trucks. The distances between the vehicles are defined by the parameter S and are 4, 8, and 12 m. The described model is an example of the study of the flow around a body in a closed tunnel. Performing numerical calculations using the finite volume method requires the definition of boundary conditions on the surfaces of the tunnel and individual vehicles.

The structure of the calculation model is shown in Figure 4. Using the symmetry of the system, only half of the domain is modeled and the symmetry condition is applied on the surface y = 0 (Surface F). The road (Surface A) and the surfaces of the trucks (Surfaces

$G_1, G_2, G_3$) have a no-slip condition. Additionally, the bottom plane (Surface A) is defined as a moving wall with a velocity of 25 m/s in the x direction. The inlet condition is applied at the front of the channel on Surface B. The value of the air velocity is 25 m/s. An outlet boundary condition is defined on the opposite wall (Surface E). On the Surfaces C and D, the zero gradient condition of all quantities is selected. The blocks are connected with each other by shared surfaces (Surfaces $H_1, H_2, ..., H_{12}$). The structure of the grids on these surfaces is the same. The values of all quantities are also the same. A detailed description of the boundary conditions of all surfaces is provided in Equations (6)–(31).

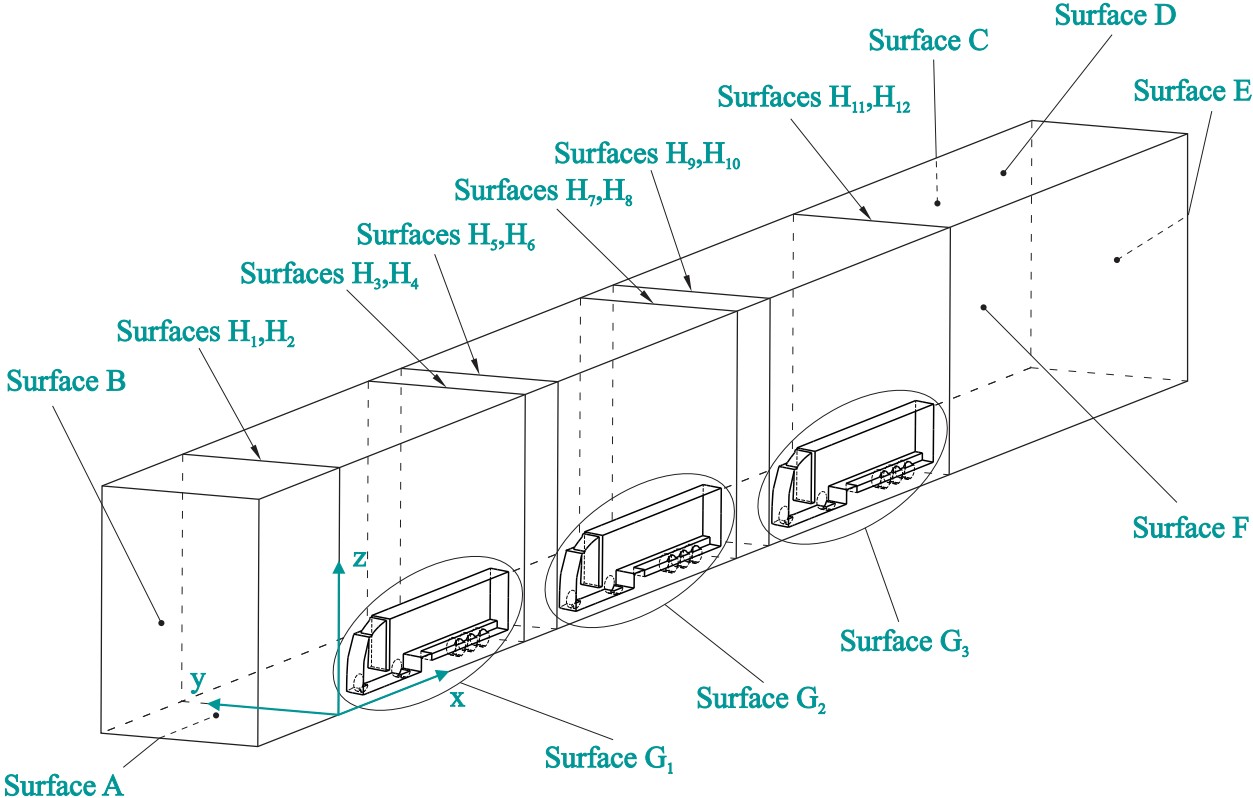

**Figure 4.** Surface description for boundary conditions in the example of a truck platoon with spacing of 8 m.

(Surface A) The bottom surface of the channel is defined as a moving wall with constant velocity of 25 m/s:

$$u_x = 25\frac{m}{s}, \quad u_y = 0, \quad u_z = 0 \tag{6}$$

$$\frac{\partial k}{\partial z} = 0, \quad \frac{\partial \omega}{\partial z} = 0, \quad \nabla p = 0 \tag{7}$$

$$k_w = \frac{u_\tau^2}{\sqrt{\beta_\infty^*}} \tag{8}$$

$$\omega_w = \frac{u_\tau}{\kappa y_w \sqrt{\beta_\infty^*}} \tag{9}$$

| | | | | |
|---|---|---|---|---|
| for | $S = 4$ m: | $-25$ m $\leq x \leq 194$ m, | $0 \leq y \leq 8$ m, | $z = 0$ |
| for | $S = 8$ m: | $-25$ m $\leq x \leq 202$ m, | $0 \leq y \leq 8$ m, | $z = 0$ |
| for | $S = 12$ m: | $-25$ m $\leq x \leq 210$ m, | $0 \leq y \leq 8$ m, | $z = 0$ |

where:

$k_w$—turbulent kinetic energy in the wall cell



$\omega_w$—specific turbulence dissipation in the wall cell
$y_w$—distance from wall to cell centroid
$u_\tau$—friction velocity
$\beta_\infty^* = 0.09$—model constant
$\kappa = 0.41$—Karman constant.

(Surface B) At the inlet of the channel, the air velocity is fixed at a constant value of 25 m/s:

$$u_x = 25\frac{m}{s}, \quad u_y = 0, \quad u_z = 0 \tag{10}$$

$$\nabla p = 0 \tag{11}$$

$$k = \frac{3}{2}(u_x I)^2 \tag{12}$$

$$\omega = \rho_p \frac{k}{\mu_p}\left(\frac{\mu_t}{\mu_p}\right)^{-1} \tag{13}$$

for:   $x = -25$ m,     $0 \le y \le 8$ m,     $0 \le z \le 12$ m

where:
$I = 1\%$—turbulence intensity
$\frac{\mu_t}{\mu_p} = 10$—turbulent viscosity ratio.

(Surface C) The side of the channel is defined as a symmetry:

$$u_y = 0 \tag{14}$$

$$\frac{\partial u_x}{\partial y} = 0, \quad \frac{\partial u_y}{\partial y} = 0, \quad \frac{\partial u_z}{\partial y} = 0, \quad \frac{\partial p}{\partial y} = 0, \quad \frac{\partial k}{\partial y} = 0, \quad \frac{\partial \omega}{\partial y} = 0 \tag{15}$$

for  $S = 4$ m:     $-25$ m $\le x \le 194$ m,     $y = 8$ m,     $0 \le z \le 12$ m
for  $S = 8$ m:     $-25$ m $\le x \le 202$ m,     $y = 8$ m,     $0 \le z \le 12$ m
for  $S = 12$ m:    $-25$ m $\le x \le 210$ m,     $y = 8$ m,     $0 \le z \le 12$ m

(Surface D) The top of the channel is defined as a symmetry:

$$u_z = 0 \tag{16}$$

$$\frac{\partial u_x}{\partial z} = 0, \quad \frac{\partial u_y}{\partial z} = 0, \quad \frac{\partial u_z}{\partial z} = 0, \quad \frac{\partial p}{\partial z} = 0, \quad \frac{\partial k}{\partial z} = 0, \quad \frac{\partial \omega}{\partial z} = 0 \tag{17}$$

for  $S = 4$ m:     $-25$ m $\le x \le 194$ m,     $0 \le y \le 8$ m,     $z = 12$ m
for  $S = 8$ m:     $-25$ m $\le x \le 202$ m,     $0 \le y \le 8$ m,     $z = 12$ m
for  $S = 12$ m:    $-25$ m $\le x \le 210$ m,     $0 \le y \le 8$ m,     $z = 12$ m

(Surface E) At the outlet of the channel the pressure is fixed:

$$p = 0 \tag{18}$$

$$\nabla u = 0, \quad \nabla k = 0, \quad \nabla \omega = 0 \tag{19}$$

$$k = \frac{3}{2}(u_{avg} I)^2 \tag{20}$$

$$\omega = \rho_p \frac{k}{\mu_p}\left(\frac{\mu_t}{\mu_p}\right)^{-1} \tag{21}$$

for  $S = 4$ m:    $x = 194$ m,    $0 \le y \le 8$ m,    $0 \le z \le 12$ m
for  $S = 8$ m:    $x = 202$ m,    $0 \le y \le 8$ m,    $0 \le z \le 12$ m
for  $S = 12$ m:    $x = 210$ m,    $0 \le y \le 8$ m,    $0 \le z \le 12$ m

where:

$u_{avg}$—mean flow velocity
$I = 5\%$—backflow turbulent intensity
$\frac{\mu_t}{\mu_p} = 10$—backflow turbulent viscosity ratio.

(Surface F) The side of the channel with truck contours is defined as a symmetry:

$$u_y = 0 \tag{22}$$

$$\frac{\partial u_x}{\partial y} = 0, \quad \frac{\partial u_y}{\partial y} = 0, \quad \frac{\partial u_z}{\partial y} = 0, \quad \frac{\partial p}{\partial y} = 0, \quad \frac{\partial k}{\partial y} = 0, \quad \frac{\partial \omega}{\partial y} = 0 \tag{23}$$

for  $S = 4$ m:    $-25$ m $\le x \le 194$ m,    $y = 0$,    $0 \le z \le 12$ m
for  $S = 8$ m:    $-25$ m $\le x \le 202$ m,    $y = 0$,    $0 \le z \le 12$ m
for  $S = 12$ m:    $-25$ m $\le x \le 210$ m,    $y = 0$,    $0 \le z \le 12$ m

(Surface $G_1, G_2, G_3$) The surfaces of the trucks are defined as a wall with a no-slip condition:

$$u_x = 0, \quad u_y = 0, \quad u_z = 0 \tag{24}$$

$$\nabla k \cdot \overline{n} = 0, \quad \nabla \omega \cdot \overline{n} = 0, \quad \nabla p = 0 \tag{25}$$

$$k_w = \frac{u_\tau^2}{\sqrt{\beta_\infty^*}} \tag{26}$$

$$\omega_w = \frac{u_\tau}{\kappa y_w \sqrt{\beta_\infty^*}} \tag{27}$$

for  $S = 4m$ :
truck 1:    $2$ m $\le x \le 16$ m,    $0 \le y \le 1.24$ m,    $0 \le z \le 4$ m
truck 2:    $20$ m $\le x \le 34$ m,    $0 \le y \le 1.24$ m,    $0 \le z \le 4$ m
truck 3:    $38$ m $\le x \le 52$ m,    $0 \le y \le 1.24$ m,    $0 \le z \le 4$ m

for  $S = 8$ m:
truck 1:    $2$ m $\le x \le 16$ m,    $0 \le y \le 1.24$ m,    $0 \le z \le 4$ m
truck 2:    $24$ m $\le x \le 38$ m,    $0 \le y \le 1.24$ m,    $0 \le z \le 4$ m
truck 3:    $46$ m $\le x \le 60$ m,    $0 \le y \le 1.24$ m,    $0 \le z \le 4$ m

for  $S = 12$ m:
truck 1:    $2$ m $\le x \le 16$ m,    $0 \le y \le 1.24$ m,    $0 \le z \le 4$ m
truck 2:    $28$ m $\le x \le 42$ m,    $0 \le y \le 1.24$ m,    $0 \le z \le 4$ m
truck 3:    $54$ m $\le x \le 68$ m,    $0 \le y \le 1.24$ m,    $0 \le z \le 4$ m

where:
$\overline{n}$—unit vector normal to surface.

(Surfaces $H_1, H_2, ..., H_{12}$) Shared surfaces between blocks are defined as conformal interfaces. The following dependencies take place in the equations under consideration:
for    $(S = 4$ m $\wedge \quad d \in \{1; 11\}) \rightarrow e = 1$;    for    $(S = 4$ m $\wedge \quad d \in \{3; 7\}) \rightarrow e = 3$.

$$\text{for} \quad S = 4 \text{ m}: \quad \forall_{\substack{a,b,c \in N^+, \\ d \in \{1,3,7,11\}, \\ e \in \{1,3\}, \\ a \le a_{max}, \\ b \le b_{max}, \\ c \le c_{max}}} u_{a,b,c}^{H_d} = u_{a,b,c}^{H_{d+e}}; \quad for \quad S \in \{8; 12\} \text{ m}: \quad \forall_{\substack{a,b,c \in N^+, \\ d \in \{1,3,5,7,9,11\}, \\ a \le a_{max}, \\ b \le b_{max}, \\ c \le c_{max}}} u_{a,b,c}^{H_d} = u_{a,b,c}^{H_{d+1}} \tag{28}$$

$$\text{for} \quad S = 4\,\text{m}: \quad \forall_{\substack{a,b,c \in N^+, \\ d \in \{1,3,7,11\}, \\ e \in \{1,3\}, \\ a \leq a_{max}, \\ b \leq b_{max}, \\ c \leq c_{max}}} p_{a,b,c}^{H_d} = p_{a,b,c}^{H_{d+e}}; \quad \text{for} \quad S \in \{8;12\}\,\text{m}: \quad \forall_{\substack{a,b,c \in N^+, \\ d \in \{1,3,5,7,9,11\}, \\ a \leq a_{max}, \\ b \leq b_{max}, \\ c \leq c_{max}}} p_{a,b,c}^{H_d} = p_{a,b,c}^{H_{d+1}} \tag{29}$$

$$\text{for} \quad S = 4\,\text{m}: \quad \forall_{\substack{a,b,c \in N^+, \\ d \in \{1,3,7,11\}, \\ e \in \{1,3\}, \\ a \leq a_{max}, \\ b \leq b_{max}, \\ c \leq c_{max}}} k_{a,b,c}^{H_d} = k_{a,b,c}^{H_{d+e}}; \quad \text{for} \quad S \in \{8;12\}\,\text{m}: \quad \forall_{\substack{a,b,c \in N^+, \\ d \in \{1,3,5,7,9,11\}, \\ a \leq a_{max}, \\ b \leq b_{max}, \\ c \leq c_{max}}} k_{a,b,c}^{H_d} = k_{a,b,c}^{H_{d+1}} \tag{30}$$

$$\text{for} \quad S = 4\,\text{m}: \quad \forall_{\substack{a,b,c \in N^+, \\ d \in \{1,3,7,11\}, \\ e \in \{1,3\}, \\ a \leq a_{max}, \\ b \leq b_{max}, \\ c \leq c_{max}}} \omega_{a,b,c}^{H_d} = \omega_{a,b,c}^{H_{d+e}}; \quad \text{for} \quad S \in \{8;12\}\,\text{m}: \quad \forall_{\substack{a,b,c \in N^+, \\ d \in \{1,3,5,7,9,11\}, \\ a \leq a_{max}, \\ b \leq b_{max}, \\ c \leq c_{max}}} \omega_{a,b,c}^{H_d} = \omega_{a,b,c}^{H_{d+1}} \tag{31}$$

<div align="center">

for $S = 4$ m:

| | | | |
|---|---|---|---|
| Surfaces $H_1$, $H_2$: | $x = 0$, | $0 \leq y \leq 8$ m, | $0 \leq z \leq 12$ m |
| Surfaces $H_3$, $H_6$: | $x = 18$ m, | $0 \leq y \leq 8$ m, | $0 \leq z \leq 12$ m |
| Surfaces $H_7$, $H_{10}$: | $x = 36$ m, | $0 \leq y \leq 8$ m, | $0 \leq z \leq 12$ m |
| Surfaces $H_{11}$, $H_{12}$: | $x = 54$ m, | $0 \leq y \leq 8$ m, | $0 \leq z \leq 12$ m |

for $S = 8$ m:

| | | | |
|---|---|---|---|
| Surfaces $H_1$, $H_2$: | $x = 0$, | $0 \leq y \leq 8$ m, | $0 \leq z \leq 12$ m |
| Surfaces $H_3$, $H_4$: | $x = 18$ m, | $0 \leq y \leq 8$ m, | $0 \leq z \leq 12$ m |
| Surfaces $H_5$, $H_6$: | $x = 22$ m, | $0 \leq y \leq 8$ m, | $0 \leq z \leq 12$ m |
| Surfaces $H_7$, $H_8$: | $x = 40$ m, | $0 \leq y \leq 8$ m, | $0 \leq z \leq 12$ m |
| Surfaces $H_9$, $H_{10}$: | $x = 44$ m, | $0 \leq y \leq 8$ m, | $0 \leq z \leq 12$ m |
| Surfaces $H_{11}$, $H_{12}$: | $x = 62$ m, | $0 \leq y \leq 8$ m, | $0 \leq z \leq 12$ m |

for $S = 12$ m:

| | | | |
|---|---|---|---|
| Surfaces $H_1$, $H_2$: | $x = 0$, | $0 \leq y \leq 8$ m, | $0 \leq z \leq 12$ m |
| Surfaces $H_3$, $H_4$: | $x = 18$ m, | $0 \leq y \leq 8$ m, | $0 \leq z \leq 12$ m |
| Surfaces $H_5$, $H_6$: | $x = 26$ m, | $0 \leq y \leq 8$ m, | $0 \leq z \leq 12$ m |
| Surfaces $H_7$, $H_8$: | $x = 44$ m, | $0 \leq y \leq 8$ m, | $0 \leq z \leq 12$ m |
| Surfaces $H_9$, $H_{10}$: | $x = 52$ m, | $0 \leq y \leq 8$ m, | $0 \leq z \leq 12$ m |
| Surfaces $H_{11}$, $H_{12}$: | $x = 70$ m, | $0 \leq y \leq 8$ m, | $0 \leq z \leq 12$ m |

</div>

where:

$a$—grid index on the x axis
$b$—grid index on the y axis
$c$—grid index on the z axis
$d$—interface number
$e$—constant.

### 2.5. Initial Condition

The obtained velocity field for the steady state is used as an initial condition for calculations using the LES turbulence model (32). Applying such a strategy does not statistically affect the stationary solution, but it will help to reach this stage in a shorter simulation time.

$$\forall_{\substack{a,b,c \in N^+, \\ a \leq a_{max}, \\ b \leq b_{max}, \\ c \leq c_{max}}} u_{a,b,c}(t = 0) = u_{a,b,c}^{k-\omega}(t = \infty) \tag{32}$$

where:

$a$—grid index on the x axis

*b*—grid index on the y axis

*c*—grid index on the z axis.

## 3. Results

### 3.1. Convergence Criteria

The aim of the research is to determine the aerodynamic parameters and the acoustic field around the column of trucks. RANS models are designed for steady-state calculations and allow for the estimation of drag and lift coefficients. In order to analyze the acoustic field, it is necessary to perform a time analysis. The calculations are divided into three stages. The first two are simulations in the steady state using first (100 iterations) and second (2000 iterations) order momentum equations, respectively. The third stage is transient computation with a time step of $10^{-5}$ seconds. The convergence criteria for each of the steps are: residuals less than $10^{-3}$, residuals less than $10^{-4}$, and residuals less than $10^{-4}$. The presented number of iterations allows for the achievement of the required convergence criterion. Calculation settings and parameters of simulation are presented in Table 3.

**Table 3.** Convergence criteria, adopted parameters, and the mathematical models used for the numerical calculations.

| Stage | I | II | III |
|---|---|---|---|
| Convergence criteria | Residuals under $10^{-3}$ | Residuals under $10^{-4}$ | Residuals under $10^{-4}$ |
| Number of iterations | 100 | 2000 | 1,00,000 |
| Turbulence model | $k - \omega$ SST | $k - \omega$ SST | LES |
| Pressure equation | Second order | Second order | PRESTO |
| Momentum equation | First order | Second order | Bounded central differencing |
| Relaxation factor—pressure | 0.25 | 0.25 | 0.35 |
| Relaxation factor—momentum | 0.25 | 0.25 | 0.35 |

### 3.2. Analysis of Aerodynamic Parameters of a Homogeneous Column

The column consisting of three vehicles is an important object in the research field. The first vehicle travels in an undisturbed stream of air. An overpressure area is created at the front of the vehicle. A vacuum zone is normally created behind it, but in the case of a vehicle column, this zone is limited by the build-up of overpressure in front of vehicle number two. The opposite situation occurs for vehicle number three. The stream is disturbed by vehicles one and two. This time, there is no car behind the vehicle, so a free wake may form. When analyzing the drag force coefficient for vehicle number three, a decrease is still observed. This is due to the reduced overpressure zone at the front of the vehicle. The position in the middle of the column turns out to be the most advantageous. Both the overpressure and underpressure zones are limited. The resulting pressure difference is therefore smaller, which results in a lower drag force acting on the vehicle.

The comparison of the drag and lift force coefficients for the two turbulence models is presented in Table 4. The first column refers to the number of vehicles N. The second column shows the distance between the trucks (spacing) S. The third column shows the order of the trucks in the column. The numbering follows the direction of travel, for example, Truck 1 is the first vehicle (in the lead) in the considered model. Columns 4–7 refer to the calculated aerodynamic coefficients. For steady-state simulations using the $k - \omega$ SST turbulence model, it is the last calculated value. For the calculations in the transient state, using the LES turbulence model, an arithmetic mean in the time range from 0.1 to 1 s is taken. The differences between the values of the drag coefficient between different models do not exceed 12%.

The calculation and analysis of the results for a single vehicle are the subject of a separate article. For the purpose of comparing individual quantities, some results are presented in this study.

A detailed history of the values of the aerodynamic coefficients is shown in Figures 5–7. The data present a comparison of the drag force coefficient and the lift force coefficient for three spacings between vehicles: 4, 8, and 12 m, as a function of the number of iterations. Figure 5 applies to trucks at the front of the column. Figure 6 shows the trucks in the center of the column. Figure 7 shows the trucks at the end of the column.

For the first truck, the drag coefficient (Figure 5a) decreases with the distance between the vehicles. It is associated with the overpressure area that develops in front of truck number two. The smaller the spacing, the greater the influence of this zone. There is also a limit distance for which this zone will not form and the drag coefficient could increase [52]. In the case of lift coefficient (Figure 5b), the value is similar to the model of a single truck. The first vehicle is running in an undisturbed air stream. The influence of the second vehicle on the area of pressure under and above the first truck is negligible. The drag coefficients for the second and third trucks (Figures 6a and 7a) in the row are stable. It is the effect of the wake from the truck number one. The lift coefficient of the second truck in Figure 6b is positive. With high velocity, this could significantly affect the tire adhesion and safety. The value of the lift coefficient for trucks at the end of column (Figure 7b is neutral, so a fluctuation of this parameter around zero is observed.

**Table 4.** Drag coefficient and lift coefficient for a single truck and three homogeneous columns.

| Number of Trucks $N$ [-] | Spacing $S$ [m] | Name | $C_{d,k-\omega}$ [-] | $C_{d,LES}$ [-] | $C_{l,k-\omega}$ [-] | $C_{l,LES}$ [-] |
|---|---|---|---|---|---|---|
| 1 | - | Truck 1 | 0.572 | 0.535 | −0.127 | −0.071 |
| 3 | 4 | Truck 1 | 0.477 | 0.419 | −0.083 | −0.051 |
| | | Truck 2 | 0.346 | 0.379 | 0.107 | 0.083 |
| | | Truck 3 | 0.395 | 0.414 | 0.023 | 0.004 |
| | | Mean | 0.406 | 0.404 | 0.016 | 0.012 |
| 3 | 8 | Truck 1 | 0.529 | 0.479 | −0.095 | −0.033 |
| | | Truck 2 | 0.372 | 0.347 | 0.063 | 0.083 |
| | | Truck 3 | 0.406 | 0.392 | 0.037 | −0.007 |
| | | Mean | 0.436 | 0.406 | 0.002 | 0.015 |
| 3 | 12 | Truck 1 | 0.561 | 0.515 | −0.110 | −0.031 |
| | | Truck 2 | 0.365 | 0.344 | 0.002 | 0.058 |
| | | Truck 3 | 0.386 | 0.370 | 0.015 | 0.013 |
| | | Mean | 0.437 | 0.410 | −0.031 | 0.014 |

The instantaneous values of the quantities are presented in the plane of symmetry (Figures 8–11). The results are obtained after 1 s of airflow using the LES turbulence model. Figures 8 and 9 are related to the value of the velocity around trucks. Due to the local acceleration of the air stream and the adopted truck speed of 25 m/s, it was decided to adopt a size scale of up to 40 m/s. Figure 9 shows the velocity field distribution, while Figure 8 shows streamlines. In Figure 10, pressure distribution expressed in Pascals is shown. The limit values are set to −1000 for underpressure and 500 for the overpressure areas. In Figure 11, turbulence intensity expressed in kg · m$^{-1}$· s$^{-1}$ is shown. Each figure contains information about the four tested models: (a) is for the study of a single truck, (b) is a column of trucks with spacing of 4 m, (c) is a column of trucks with spacing of 8 m, and (d) is a column of trucks with spacing of 12 m. Due to the large size of the computing domain, reaching 200 m, it was decided to present only a fragment of the model related to the surroundings of the studied vehicles.

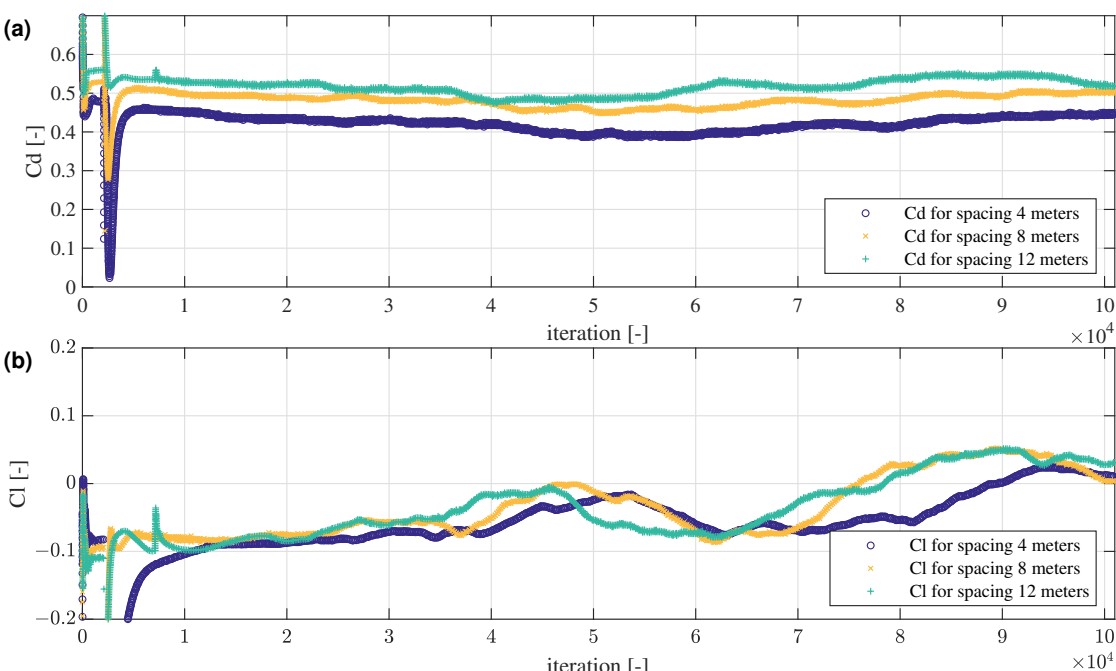

**Figure 5.** Comparison of the aerodynamic coefficients for homogeneous columns consisting of three trucks. The graph shows data for the first vehicle at the front of the fleet. Dark blue "o"—the distance between the vehicles is 4 m, orange "x"—the distance between the vehicles is 8 m, teal "+"—the distance between the vehicles is 12 m. (**a**) Drag coefficient. (**b**) Lift coefficient.

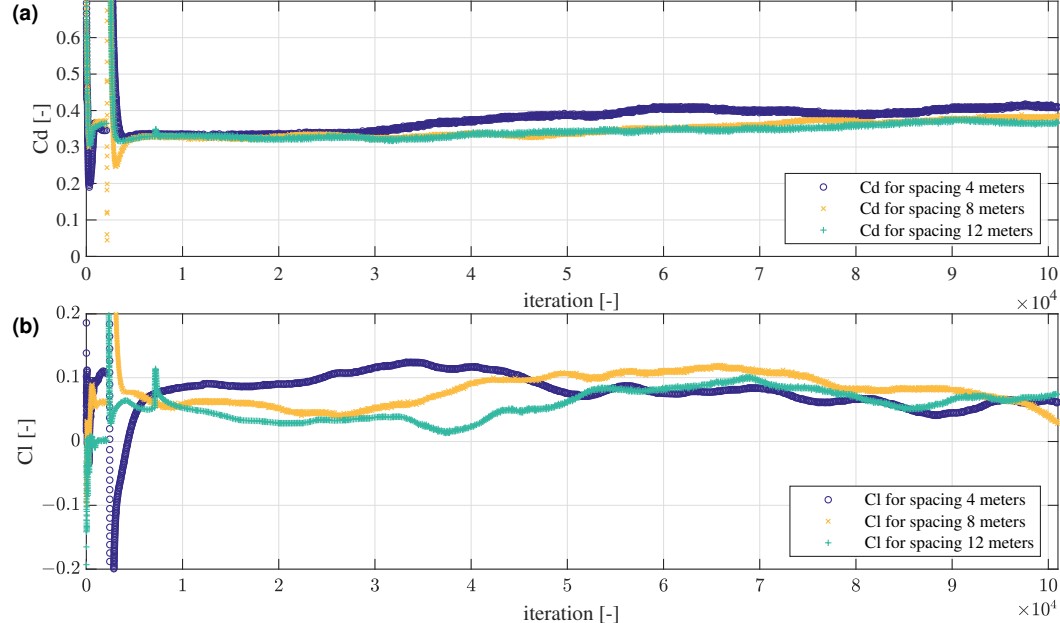

**Figure 6.** Comparison of the aerodynamic coefficients for homogeneous columns consisting of three trucks. The graph shows the data for the second vehicle in the center of the fleet. Dark blue "o"—the distance between the vehicles is 4 m, orange "x"—the distance between the vehicles is 8 m, teal "+"—the distance between the vehicles is 12 m. (**a**) Drag coefficient. (**b**) Lift coefficient.

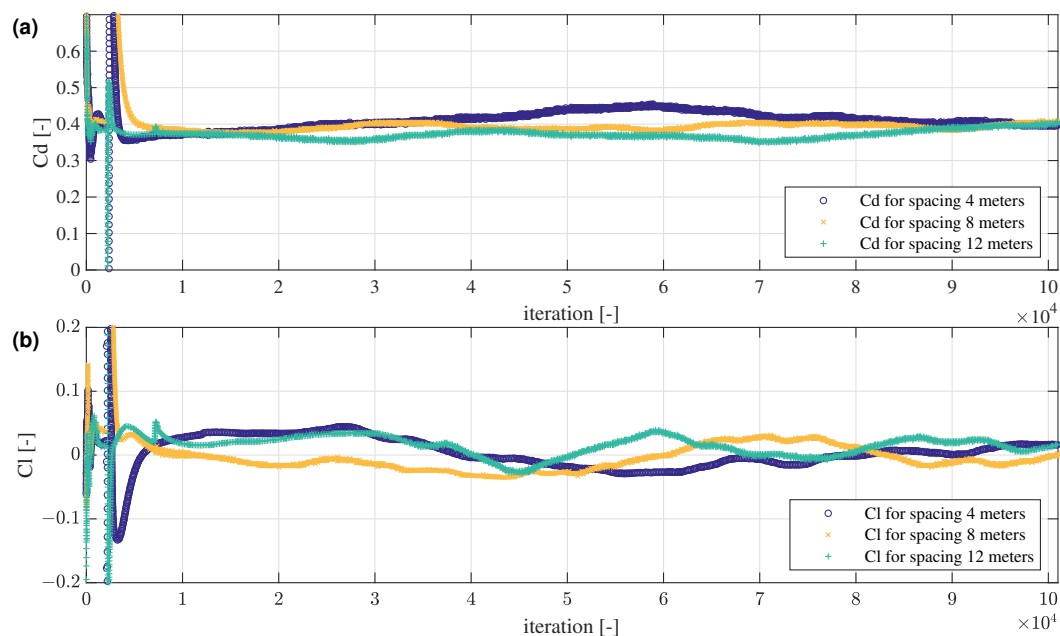

**Figure 7.** Comparison of the aerodynamic coefficients for homogeneous columns consisting of three trucks. The graph shows data for vehicle number three at the end of the fleet. Dark blue "o"—the distance between the vehicles is 4 m, orange "x"—the distance between the vehicles is 8 m, teal "+"—the distance between the vehicles is 12 m. (**a**) Drag coefficient. (**b**) Lift coefficient.

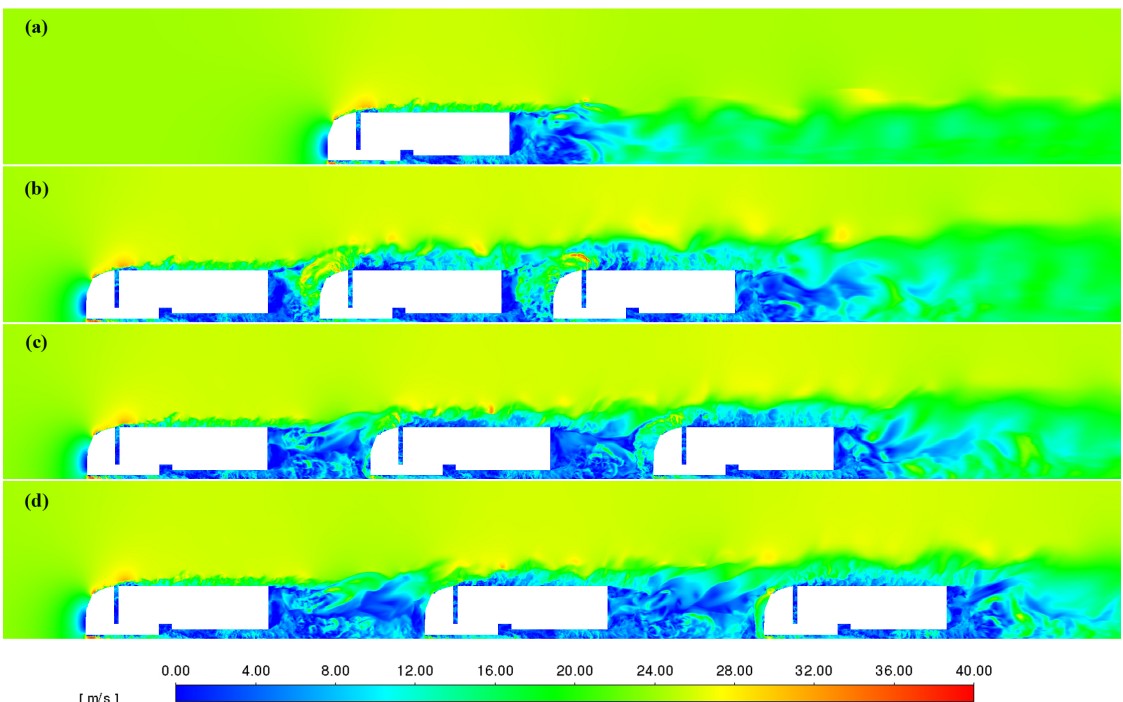

**Figure 8.** A fragment of velocity field around the trucks. View at the symmetry plane. (**a**) Single truck. (**b**) Column of trucks with spacing of 4 m. (**c**) Column of trucks with spacing of 8 m. (**d**) Column of trucks with spacing of 12 m.

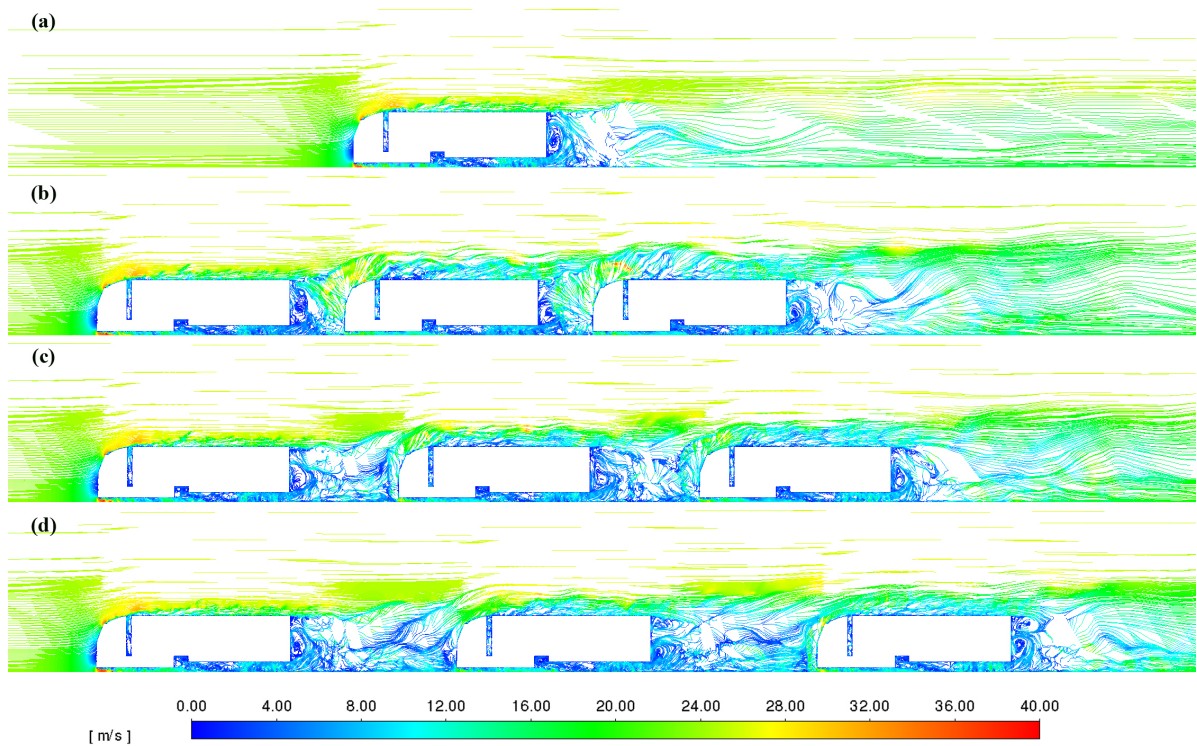

**Figure 9.** A fragment of streamlines around the trucks. View at the symmetry plane. (**a**) Single truck. (**b**) Column of trucks with spacing of 4 m. (**c**) Column of trucks with spacing of 8 m. (**d**) Column of trucks with spacing of 12 m.

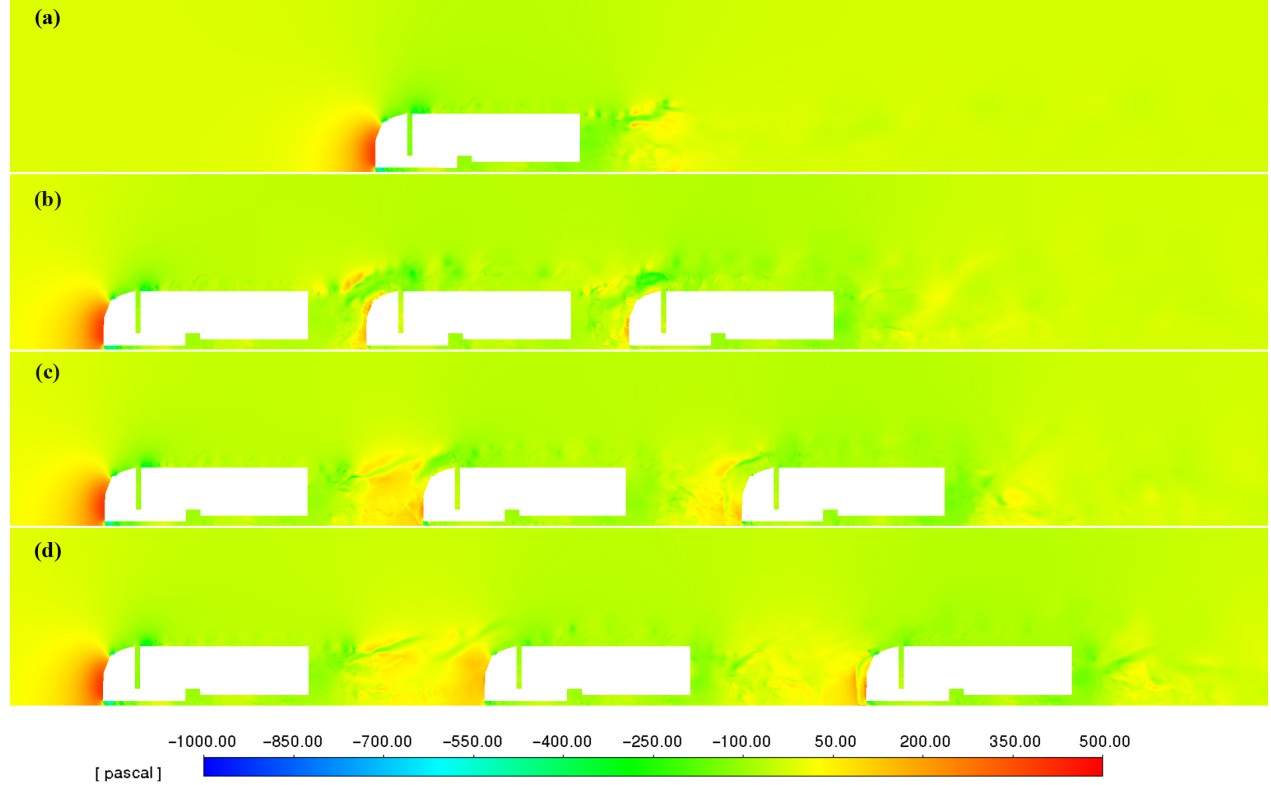

**Figure 10.** A fragment of pressure distribution around the trucks. View at the symmetry plane. (**a**) Single truck. (**b**) Column of trucks with spacing of 4 m. (**c**) Column of trucks with spacing of 8 m. (**d**) Column of trucks with spacing of 12 m.

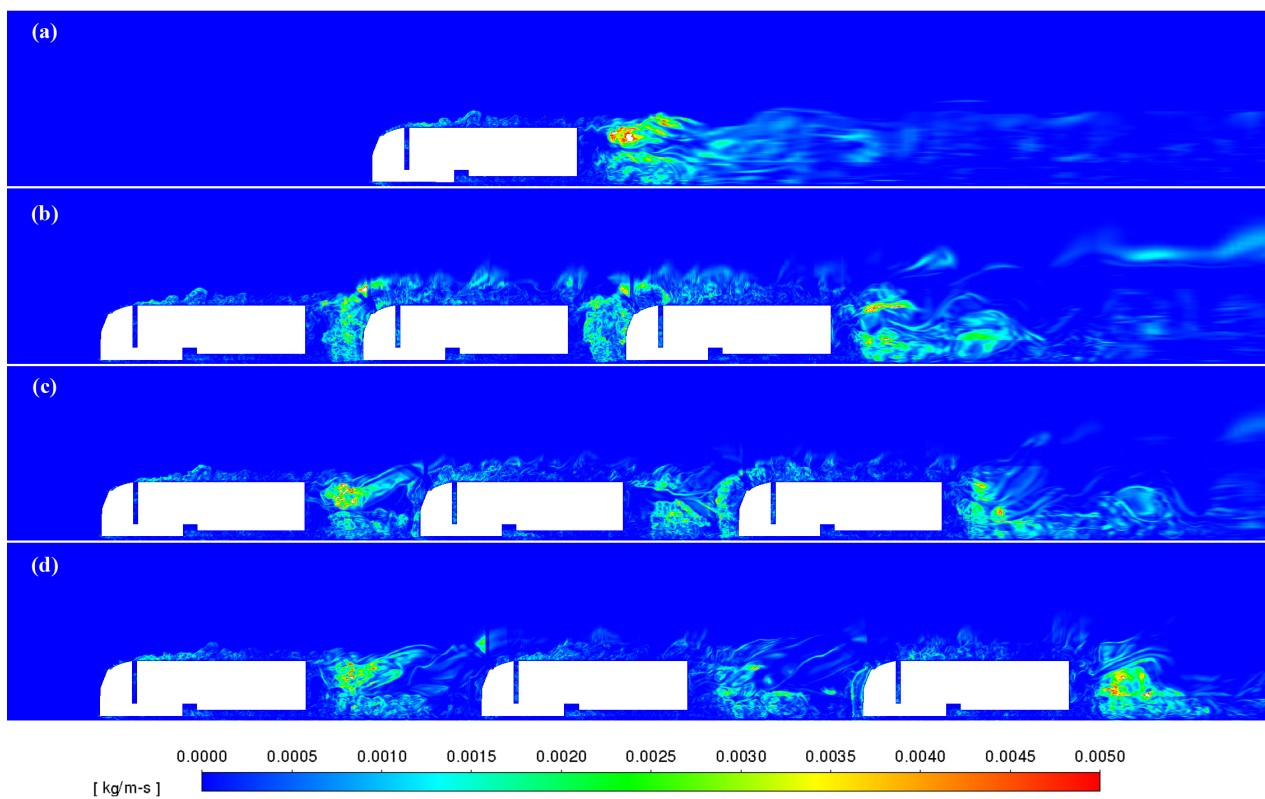

0.0000  0.0005  0.0010  0.0015  0.0020  0.0025  0.0030  0.0035  0.0040  0.0045  0.0050

[ kg/m-s ]

**Figure 11.** A fragment of turbulence intensity around the trucks. View at the symmetry plane. (**a**) Single truck. (**b**) Column of trucks with spacing of 4 m. (**c**) Column of trucks with spacing of 8 m. (**d**) Column of trucks with spacing of 12 m.

### 3.3. Analysis of Acoustic Field around a Homogeneous Column

Figures 12–15 show the top view of a column of truck with the value of the overall sound pressure level (OASPL). The OASPL was calculated according to Equation (4), at discrete points distanced from the side surfaces of the vehicles by 1, 2, and 4 m, respectively. The receivers are located at the beginning, middle, and end of each truck, at a height of 1.7 m. The number inside the data point is the receiver number. The OASPL value is shown in the upper left corner above the data point. The value of the OASPL is also presented using a color bar. Parameter p' in the equation is substituted by the root mean square pressure fluctuation from the time range of 0.1 to 1 s.

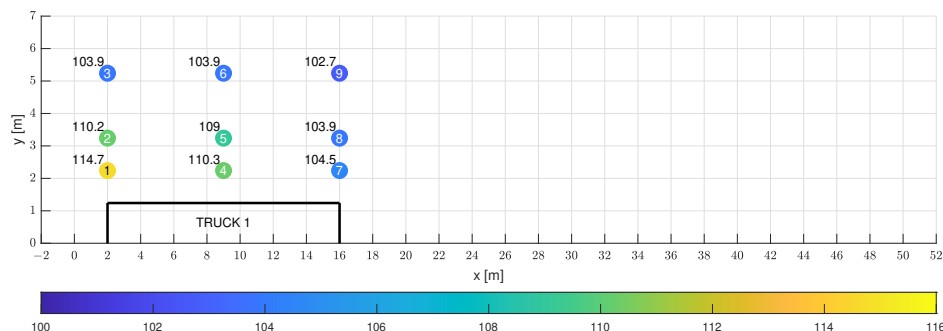

100   102   104   106   108   110   112   114   116

**Figure 12.** Overall sound pressure level in decibels (dB) on the side of the single truck, measured by receivers for the z-coordinate of 1.7 m. View from the top.

For a single truck, the highest values of the overall sound pressure level were observed in points 1, 2, 3, that is, in the first cross-section, while the lowest were found in the third cross-section, that is, in points 7, 8, 9. The maximum value of the overall sound pressure level was 114.7 dB at point 1, i.e., at a distance of 1 m from the vehicle, and the minimum

value of 102.7 dB at point 9, at a distance of 4 m from the truck. Additionally, the largest difference in overall sound pressure levels was observed between points 1 and 3 and was 10.8 dB and the smallest was 1.8 dB between points 7 and 8.

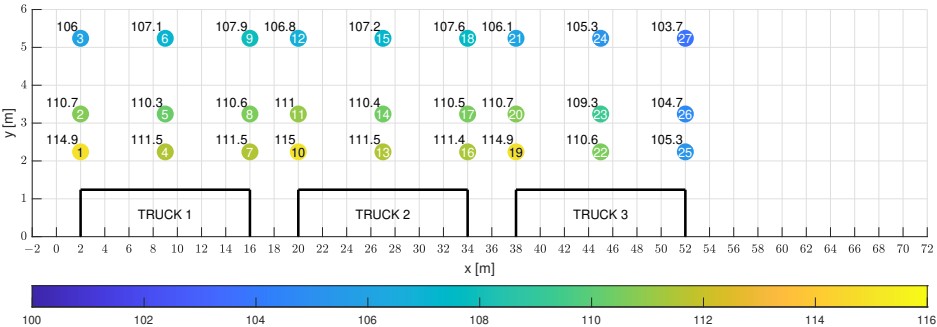

**Figure 13.** Overall sound pressure level in decibels (dB) on the side of the trucks, measured by receivers for the z-coordinate of 1.7 m. Spacing between trucks: 4 m. View from the top.

On the basis of the analysis of the results of the presented distributions of the overall sound pressure level for three columns of trucks, we can state that at a distance of 1 m from each of the vehicles, i.e., at points 1, 10, 19, in all cases, the highest levels occur. The lowest levels of the overall sound pressure level were observed in the last cross-section, i.e., at points 25, 26, 27. The biggest differences in observed an overall sound pressure levels, similarly to what was observed for a single truck, occur at points 1 and 3 and amount to 8.9 dB for a 4 m distance between vehicles, 10.1 dB for an 8 m distance between vehicles, and 9.4 dB for a 12 m distance between vehicles. This is consistent with what was found for a single truck. In addition, large differences in overall sound pressure levels are found at points 10 and 12, and 19 and 21, for all three vehicle spacing distances. The smallest differences in observed an overall sound pressure levels are similar to what was found for the single truck, at points 25 and 27, and are 1.6 dB for a 4 m vehicle spacing, 1.7 dB for an 8 m vehicle spacing, and 1.8 dB for a 12 m vehicle spacing. Because of the significant differences between the overall sound pressure levels at points located in close proximity to vehicles, it was decided to perform additional calculations at a distance of 15 m and a height of 4 m and determine A-weighted sound pressure level values. The results that were obtained are presented later in this paper.

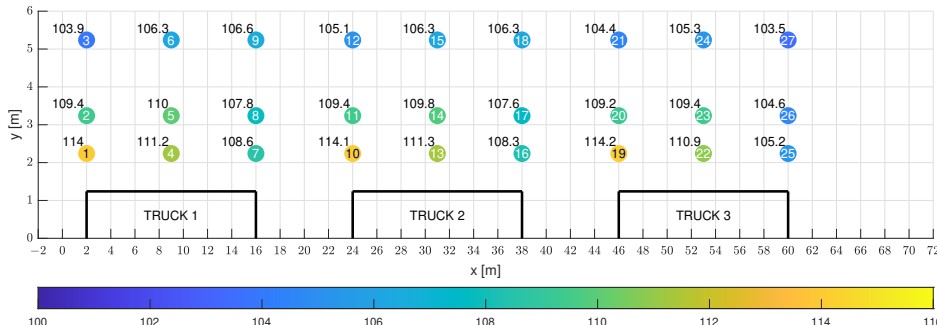

**Figure 14.** Overall sound pressure level in decibels (dB) on the side of the trucks, measured by receivers for the z-coordinate of 1.7 m. Spacing between trucks: 8 m. View from the top.

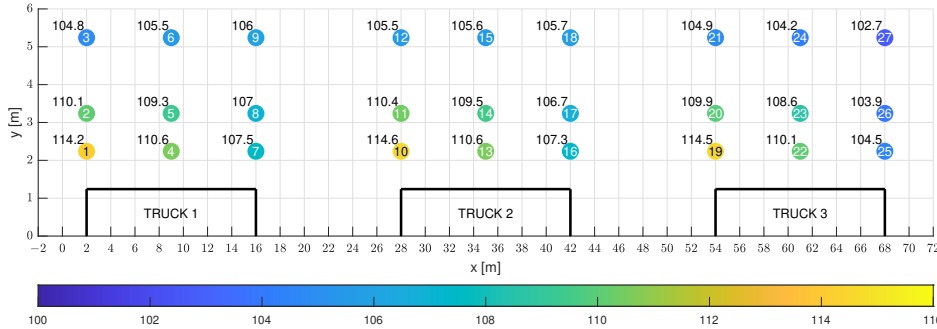

**Figure 15.** Overall sound pressure level in decibels (dB) on the side of the trucks, measured by receivers for the z-coordinate of 1.7 m. Spacing between trucks: 12 m. View from the top.

### 3.4. Field Measurements and Computational Verification

In order to confirm correctness of the calculations performed, acoustic measurements were carried out on the Krakow bypass road. The measurements were made at night time at 11 p.m. The test object was a Mercedes truck moving at a speed of 90 km/h. The measurements were made with a Svan 945 sound level meter made by Svantek in windless weather and a temperature of 15 °C. The measuring point was located 15 m from the road and at a height of 4 m. In addition to measuring the equivalent sound level A for a truck passing at a speed of 90 km/h, the background noise was also measured. The test results in the form of spectra in octave bands and the A level are presented in Figure 16.

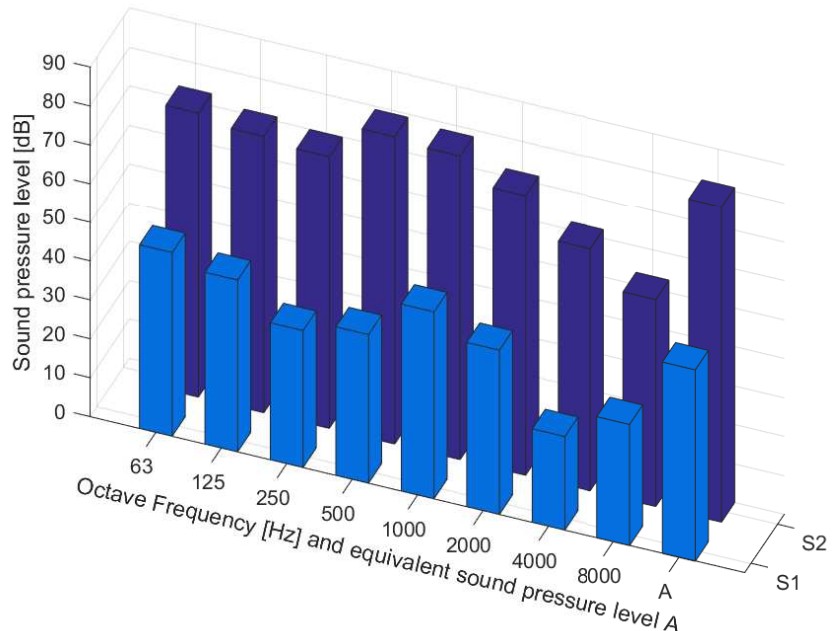

**Figure 16.** Sound pressure level in octave bands and equivalent sound pressure level A: dark blue—truck crossing, blue—acoustic background.

After the field tests were performed, the equivalent levels were determined for the developed numerical models presented previously. Due to the fact that the acoustic calculations were conducted in time using the finite volume method, the values of the equivalent sound level A were determined for the distance of 15 m and height of 4 m. For a single truck, the result was $L_{Aeq}$ −81.5 dB. For three trucks moving in a column and a distance between the vehicles of 12 m, the $L_{Aeq}$ value was 86 dB. For a column of trucks and a distance of 8 m, the $L_{Aeq}$ value was 85.9 dB, and for trucks moving in a column and a distance between vehicles of 4 m, the $L_{Aeq}$ was 86.5 dB. Comparing the equivalent sound level A results recorded for a passing single truck with the equivalent

sound level A value determined from the CFD results, it should be noted that the results are identical. In order to confirm the obtained results and due to the fact that the authors did not have three trucks at their disposal to perform field measurements, additional calculation methods recommended for performing assessments of road noise's impact on the environment were used [53,54]. Calculations were performed using SoundPlan software. At this stage of the work, using the NMPB-Routes-2008 method and ISO 9613-2, four models were developed: a single truck model and vehicle columns with different distances. The following assumptions were made in the study, which were derived from the dimensions of the trucks and the distances between them, which were then converted to the number of vehicles per hour. A vehicle speed of 90 km/h was assumed in all cases. For a column of trucks with distances of 4 m, it was calculated that there would be 5000 veh/h in the model, for 8 m 4090 veh/h, and for 12 m 3461 veh/h. After entering the data into the model, the calculations were made and the results of the equivalent sound level A were obtained; for a single truck $L_{Aeq}$ −81.6 dB, for a column of trucks with distances of 4 m $L_{Aeq}$ −86.1 dB, for 8 m $L_{Aeq}$ −85.2 dB, and for 12 m $L_{Aeq}$ −84.5 dB. When analyzing the obtained results, it should be stated that the differences between the values for the columns of trucks determined on the basis of two calculation methods, i.e., the finite volume method and the French method, are within 0.1 to 1.5 dB which is a good result. For a single truck, the results of equivalent sound level A differs by 0.1 dB.

## 4. Discussion and Conclusions

This paper presents original aerodynamic and aeroacoustic results for a single truck and identical truck models comprising a truck column traveling at 90 km/h.

In the modeling process, a geometric model of a truck and then columns of trucks were made with three different spacings between vehicles. The research process successively discretized the computational area, presented the model, and presented the boundary and initial conditions adopted in the computations. In the next stage of the work, calculations were carried out and a number of analyses were performed, including: drag and lift force coefficients and total sound pressure level distributions. Numerous illustrations are also presented: velocity, pressure, and turbulence intensity distributions around the analyzed columns of trucks.

Acoustic measurements have been made during the passage of a single Mercedes truck and additional acoustic calculations have been performed using the recommended French methodology NMPB-Routes-2008 and ISO 9613-2. Calculations were performed using SoundPlan software.

The process of verification of the developed models was carried out in three stages. In the first one, defining the convergence criterion, the highest quality of mesh was taken care of. The meshes produced were also the result of a compromise between the number of elements generated and the time required to perform the calculations. In the second stage of model validation, the differences between the obtained values of the drag coefficient for the computer model and the real vehicle were checked. The last stage of verification work included field acoustic measurements and additional acoustic calculations for identical truck models.

The presented results confirm the benefits of platooning. Moving vehicles in specially organized columns reduces the total air drag forces acting on the vehicles. This translates into lower fuel consumption and a reduction in the amount of carbon dioxide emitted into the environment. In the case of a distance between vehicles of 4 m, the reduction in the drag coefficient reached almost 30%.

Maintaining small distances between vehicles is associated with a shorter reaction time, therefore it is necessary in this case to introduce appropriate communication and control systems, such as vehicle to vehicle. Systems of this type are classified as intelligent control systems. They enable simultaneous acceleration and braking of all vehicles in the column.

For the selected three distances between vehicles, 4, 8, and 12 m, small differences in the average drag coefficients were observed. In the case of calculations for the steady state, this difference was approximately 7%, while for the transient, the average value did not exceed 1.5%. Consequently, by introducing intelligent systems such as V2V to roads, a greater distance could be used to improve safety. In future studies, it is also planned to check the distances of 25, 50, and 100 m.

Using two different turbulence models, the difference in the obtained coefficients of the drag force did not exceed 12%. In order to reduce the difference between the results, the time step should be increased. This would allow a greater spread of the real flow time while maintaining the same number of iterations. In the case of this work, it would involve a reduction in the considered frequency range for the conducted spectral analysis. On the other hand, increasing the number of iterations while keeping the time step would drastically increase the demand for computing resources. The obtained results were considered satisfactory.

The considered truck model is characterized by a drag coefficient of 0.572 ($k - \omega$ SST model) and a lift coefficient of $-0.127$ ($k - \omega$ SST model). In the case of a homogeneous column consisting of three trucks, the lowest value of the drag coefficient is observed for the truck in the middle. The reduction in this parameter is almost 40%. The situation with the lift coefficient is different. For a single truck, it is negative, which means that downforce is applied to the vehicle. This improves the vehicle's adhesion to the road, which impacts on increasing driving safety. For vehicles in the center of the column, the direction of this force changes, resulting in a reduction in the adhesion of the tires to the road. In this case, an additional analysis of the acting lift force would have to be carried out in order to determine its distribution along the length of the vehicle. For the leading truck, the lift coefficient is similar to the coefficient of the single truck.

The use of NMPB-Routes-2008 and ISO 9613-2 methods enabled a direct comparison of the simulation results obtained with the finite volume methods and acoustic measurements conducted in the field.

Analyzing the results obtained for the truck columns, it should be stated that the differences between the values of the equivalent sound level A determined on the basis of two calculation methods, i.e., the finite volume method and the French method, are within the range of 0.1 to 1.5 dB. This is a good result despite the simplifications adopted in the models. For a single truck, the results of the equivalent sound level A differ by 0.1 dB.

The obtained results presented in the paper should be considered as correct. Nevertheless, future studies will concern sensitivity analysis of the model, taking into account the influence of monopole, dipole and quadrupole sources. Numerical calculation and verification will be preceded by the implementation of, among others, the method proposed in the paper [55].

The results presented in this paper have a universal character and can be used to build intelligent transport systems (ITSs) and intelligent environmental management systems (IEMSs) for municipalities, counties, cities, and urban agglomerations by taking into account the platooning process.

**Author Contributions:** Conceptualization, W.M.H. and W.B.C.; methodology, W.M.H. and W.B.C.; software, W.M.H. and W.B.C.; validation, W.M.H. and W.B.C.; formal analysis, W.M.H.; investigation, W.M.H. and W.B.C.; writing—original draft preparation, W.M.H. and W.B.C.; review, W.B.C.; visualization, W.M.H. All authors have read and agreed to the published version of the manuscript.

**Funding:** This research received no external funding.

**Institutional Review Board Statement:** Not applicable.

**Informed Consent Statement:** Not applicable.

**Data Availability Statement:** Not applicable.

**Acknowledgments:** This research was conducted with support of the PL-Grid (Polish NGI) Consortium, coordinated by the ACC Cyfronet AGH, grant ID: plgphdaia. Simulations were performed on the supercomputer Prometheus in the Academic Computer Centre Cyfronet AGH. The third stage of studies related to transient calculation required the use of significant computing resources. The used computational wall time exceeded 2,400,000 h. In the presented work the transient calculations were necessary due to the need to perform spectral analysis. In other cases, if only aerodynamic parameters such as drag or lift coefficients were needed, the steady-state simulation was sufficient. Each numerical case was calculated on 10 nodes with a total number of cores of 240. Because of the size of the model, the needed computer memory RAM for post-processing exceeded 600GB. The geometry models were created in CATIA v5 software. Mesh preparation was conducted in ANSYS ICEM. As a solver, ANSYS Fluent was used. Fast Fourier transform and further calculation and visualizations were made in MATLAB.

**Conflicts of Interest:** The authors declare no conflict of interest.

## Abbreviations

The following abbreviations are used in this manuscript:

| | |
|---|---|
| SST | Shear Stress Transport |
| LES | Large Eddy Simulation |
| FW-H | Ffowcs Williams–Hawkings |
| ITS | Intelligent Transport System |
| SPL | Sound Pressure Level |
| RANS | Reynolds-Averaged Navier–Stokes |
| CAA | Computational Aeroacoustics |
| FFT | Fast Fourier Transformation |
| V2V | Vehicle to Vehicle |
| SARTRE | Safe Road Trains for the Environment |
| GCDC | Grand Cooperative Driving Challenge |
| ETPC | European Truck Platoon Challenge |
| DART | Dynamic Autonomous Road Transit |
| DES | Detached Eddy Simulation |
| CACC | Cooperative Adaptive Cruise Control |
| ACC | Adaptive Cruise Control |
| IEMS | Intelligent Environmental Management System |
| PRESTO | Pressure Staggering Option |
| OASPL | Overall Sound Pressure Level |

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
