# Peer review of "Numerical Analysis of Aeroacoustic Phenomena Generated by Truck Platoons"

_sustainability, doi:10.3390/su132414073_

Round 1

Reviewer 1 Report

Excellent work !!  Well-Done.

The paper is one of the most comprehensive and timely articles I've seen in recent years.  comprehensive, because the paper presents a complete analysis of aerodynamics and aeroacoustics studies of multiple trucks with variable distance between them.  The number of trucks variable as well as the distance variable are both realistic and represent real life conditions.  The paper is also timely, because there is great deal of research taking place on the subject of CAVs (Connected and Autonomous Vehicles).  But, a large majority of published articles only discuss the benefits of trucks platooning for reducing the environmental pollution, reducing fuel costs and improving the efficiency of logistics in the transportation industry.  I have not seen any articles that present such comprehensive analysis of the aerodynamics and aeroacoustics impacts of multiple trucks platooning.  The analyses the authors have presented are solid and their conclusions will have many direct applications for both governments at different levels and private transportation sectors.

Reviewer 2 Report

The manuscript entitled "Numerical Analysis of Aeroacoustic Phenomena Generated by Truck Platoons" is looking into benefits of platooning via numerical modeling. The research has practical importance on the environment. Following are major comments to authors for their consideration to review the manuscript.

  1. In general, it is a well-written manuscript, but the length is too much to follow. Therefore, the authors should consider removing some of the routine explanations and some of the redundant results from the main manuscript and share them in the appendix.
  2. It is not clear to this reviewer how the sound field distributions are calculated using Ffowcs Williams-Hawkings (FW-H) analogy? has it been added as a user-defined function (UDF) in Ansys? 
  3. In the title of Figures 13-17, the reviewer is confused if the sound pressure is measured or predicted? Whereas line 382 says it is calculated?
  4. Figures 1-4. do we really need so many images to illustrate the geometry and mesh?
  5. Discussion about boundary conditions in section 2.4 is too lengthy and confusing. The first question is do we really need all these conditions explained in the main manuscript? Perhaps a thorough job is needed in putting them as a table. 
  6. Table 2 and the convergence criterion is discussed can be abridged. I am assuming these details can be presented in a thesis, but, not in a journal paper. 
  7. Acknowledgements are part of your conclusions. For instance, from line 520 onwards, it can be separated as a section. 

Reviewer 3 Report

Review report: Numerical Analysis of Aeroacoustic Phenomena Generated by Truck Platoons
Authors: Władysław Marek Hamiga et al.

The authors investigated the aerodynamic forces and sound generated by identical truck column models. A truck model and three columns of identical trucks with different distances between the vehicles were made and tested using commercial software. The drag coefficients for each set of vehicles were reported. Then, the sound generated by moving vehicles are determined by solving the Ffowcs Williams-Hawkings equation. In order to verify the correctness of the results, field tests were also performed and additional acoustic calculations were carried out using the NMPB-Routes-2008 and ISO 9613-2 models. Calculations were performed using SoundPlan software. The performed tests showed good quality of the built aerodynamic and aeroacoustics models. 

Significant benefits of platooning for the environment have been reported. The topic is interesting and within the scope of Sustainability. The results presented in this paper have a universal character and can be used to build intelligent transport systems and intelligent environmental management systems for municipalities, counties, cities and urban agglomerations taking into account the platooning process.
The manuscript is generally well written. However, I think the concerns as follows should be addressed to improve the quality of the paper. 

1. The authors stated that "Special attention is given to the discretization process, which is realized in the ANSYS ICEM tool" (lines 128-129). However, I have not found the detailed "special attention" given in Section 2. The discussion of the "special attention" in details should be helpful to improve the novelty of the paper. 

2. The authors are suggested to discuss the differences and relations between the air velocity vector U (line 152), fluid velocity component u_i (line 156), and relative air velocity u (line 179).

3. The "Lighthill turbulence stress tensor" in line 160 is usually name as "Lighthill stress tensor". 

4. The "compressible stress tensor" in line 161 should be "compressive stress tensor"

5. The authors stated that "In presented model flow is incompressible, therefore it is possible to assume p_0 =
c^2_0 (rho - rho_0) = c_0 rho^\prime" (Lines 167-168). First, the assumption “p_0 =
c^2_0 (rho - rho_0)” is not because the flow is incompressible. Second, the "c_0 rho^\prime" on the r.h.s of the euqation should be "c^2_0 rho^\prime"  

6. The author have not report the method in solving the Ffowcs Williams-Hawkings equation. It seems that only the dipole source terms are taken into account. The recent researches that the quadrupole terms might also play an ignorable role even at low Mach number when the wake are complex, such as that discussed in details in recent work of AIAA Journal 2021, 59(11): 4809-4814 and Theoretical and Applied Mechanics Letters 2021, 11(4): 100259. The authors are suggested to discuss this point.   

7. What are "\beta^*_\inft" in Eq.(8) and "k" in Eq.(9)? The authors should give explanations. 

Round 2

Reviewer 2 Report

None

Reviewer 3 Report

While the authors partly addressed my concern, I still concern about the results of the present work. The details are as follows.

  1. The authors provided some general information on the discretization. However, the information is not enough to show the result is valid. For example, the authors stated that “The boundary layer around the vehicle consists of 15 elements in height. The height of the first element, expressed by the dimensionless y+ factor, does not exceed 1”. The authors should at least give the schematic of the local mesh near the boundary, the typical height of the first element, the way to stretch the mesh, the reason to use 15 elements within the boundary layer, and the validation to show the numerical result on this type of mesh is valid.
  2. Addressed
  3. The response to point 3 should be to point 4
  4. The response to point 4 should be to point 3
  5. I am happy to see the author correct the wrong formulation
  6. The low Mach number used in this work cannot ensure the validity of ignoring the quadrupole source, since the truck is a bluff body with strong wakes. The authors should at least mention the possible problem of the present results and the possible way to solve this problem.

Round 3

Reviewer 3 Report

The authors have addressed my concerns. 

There are some typos in line 528 "monopoly, dipole and quadropol"

Author Response

Typos in line 528 have been corrected.

Thank you for your interest and taking the time to review the submitted article.